# Lupus Anticoagulant Positivity as a Risk Marker for Hemolytic Anemia in Patients with APS

**DOI:** 10.3390/medicina61081364

**Published:** 2025-07-28

**Authors:** Ji-Hyoun Kang

**Affiliations:** Division of Rheumatology, Department of Internal Medicine, Chonnam National University Medical School & Hospital, 42 Jebong-ro, Dong-gu, Gwangju 61469, Republic of Korea; romi918@naver.com; Tel.: +82-62-670-9491

**Keywords:** antiphospholipid syndrome, hemolytic anemia, lupus anticoagulant

## Abstract

*Background and Objectives*: Thrombocytopenia and hemolytic anemia are common but non-criteria manifestations of antiphospholipid syndrome (APS). However, their relationship with specific immunological profiles remains poorly characterized. This study aimed to evaluate these hematologic manifestations and identify their serological associations in patients with APS. *Materials and Methods*: We retrospectively reviewed 346 patients diagnosed with APS. Demographic, clinical, and laboratory characteristics were analyzed. Logistic regression was used to identify risk factors associated with hemolytic anemia. *Results*: The mean age was 47.1 ± 13.1 years, and 71.7% were female. Thrombocytopenia was present in 34.5%, and hemolytic anemia in 16.5% of patients. Lupus anticoagulant (LAC) was the most common antibody (66.8%). In univariate analysis, hemolytic anemia was significantly associated with LAC positivity (OR 4.216, 95% CI: 2.326–7.640, *p* < 0.001), anticardiolipin IgG (OR 7.170, *p* = 0.007), triple positivity (OR 3.638, *p* = 0.002), and diabetes mellitus (OR 2.084, *p* = 0.007). DIAPS showed a protective trend (OR 0.547, *p* = 0.002). In multivariate analysis, only LAC remained an independent risk factor for hemolytic anemia (adjusted OR 3.557, 95% CI: 1.355–9.335, *p* = 0.003). *Conclusions*: LAC positivity is an independent predictor of hemolytic anemia in APS. These findings suggest a distinct immunologic profile among patients with hematologic involvement and highlight the need for further investigation into non-criteria manifestations.

## 1. Introduction

Antiphospholipid syndrome (APS) is a complex systemic autoimmune disorder characterized by vascular thrombosis and/or pregnancy-related complications associated with persistently positive antiphospholipid antibodies (aPLs). The most identified aPLs include lupus anticoagulant (LAC), anticardiolipin, and anti-β2 glycoprotein I antibodies, which play a central role in the pathogenesis and diagnosis of the syndrome [1,2]. Although thrombotic and obstetric complications have traditionally formed the cornerstone of APS classification, the updated 2023 ACR/EULAR criteria also incorporate non-thrombotic manifestations such as thrombocytopenia, microvascular involvement, and heart valve disease, reflecting the syndrome’s broader clinical spectrum. However, an increasing body of literature has highlighted a broader spectrum of clinical manifestations that fall outside this formal diagnostic framework. These non-criteria manifestations, though clinically significant, remain under-recognized and inadequately characterized in routine practice. The limited focus on these features may stem in part from the emphasis placed on thrombotic risk stratification and management of pregnancy-related complications, which are more immediately life-threatening and thus more heavily prioritized in routine care. Importantly, the clinical significance and underlying immunopathogenesis of the hematological manifestations of APS remain poorly defined. Thrombocytopenia in APS is often moderate and chronic, whereas hemolytic anemia tends to present in a Coombs-positive pattern, suggesting an autoimmune basis. These findings raise important questions regarding their relationship to specific aPL profiles, including the isotype, titer, and persistence of autoantibodies [1,3]. 

Previous research has primarily focused on thrombotic outcomes, often categorizing patients according to the presence of triple positivity, which refers to the simultaneous detection of all three major aPL types as strong predictors of thrombotic risk. However, the association between distinct aPL subtypes and non-thrombotic features, particularly hematological abnormalities, remains unclear [4,5,6]. Therefore, comprehensive studies examining the immunological correlates of these non-criteria features to provide better risk stratification and holistic patient management are warranted [7]. Furthermore, recent studies have suggested that certain non-criteria features may serve as early indicators of systemic autoimmunity or evolving APS phenotypes, particularly in patients with incomplete or seronegative presentation. As such, a deeper understanding of the hematologic spectrum of APS may not only enhance diagnostic sensitivity but also provide insight into disease mechanisms and guide treatment decisions. In particular, elucidating the role of LACs and other aPL subtypes in the development of hemolytic anemia could reveal key pathogenic pathways and identify high-risk patients who may benefit from closer monitoring or targeted therapeutic interventions. Notably, Bernardoff et al. [2] highlighted an association between antiphospholipid antibodies and autoimmune hemolytic anemia in patients with systemic lupus erythematosus, underscoring the plausibility of this link in APS as well. This suggests a potential pathophysiological overlap wherein LAC may contribute to red blood cell destruction via complement activation or Fc receptor-mediated clearance mechanisms.

Here, we investigate the prevalence and clinical relevance of hematological abnormalities, specifically thrombocytopenia and Coombs-positive hemolytic anemia, in a well-characterized cohort of patients with confirmed APS. Using real-world data accumulated over a 10-year observation period, we aimed to quantify the burden of these non-criteria features and explore their serological associations with various aPL subtypes. Through this approach, we hope to contribute to a more nuanced understanding of the immune-hematologic profile of APS and provide a foundation for future research aimed at refining the clinical definitions, prognostic models, and therapeutic strategies for this complex autoimmune condition.

## 2. Materials and Methods

This retrospective observational study investigated the prevalence and characteristics of hematological abnormalities in patients diagnosed with APS. The medical records of all patients who fulfilled the diagnostic criteria for APS and were treated at the Chonnam National University Hospital between March 2015 and March 2025 were systematically reviewed. Overall, 346 patients were included in this analysis. These individuals had a confirmed diagnosis of definite APS based on the revised Sapporo criteria, also known as the Sydney classification criteria, which requires the presence of at least one clinical event combined with persistently positive aPLs on two or more occasions at least 12 weeks apart [8]. However, we acknowledge the publication of the 2023 ACR/EULAR classification criteria [9], which incorporate thrombocytopenia and non-thrombotic features, and we have reflected on their implications in the Discussion section. The recent 2023 ACR/EULAR classification criteria for APS include thrombocytopenia, heart valve disease, and microangiopathy as part of the non-thrombotic manifestations. Our findings, particularly the association between thrombocytopenia and autoimmune hemolytic anemia (AIHA), are consistent with this broadened view of APS. Although our study used the 2006 Sydney criteria due to its retrospective nature, the results support the relevance of hematologic features in future APS stratification frameworks. Given the retrospective nature of the study and the use of anonymized data extracted from existing electronic medical records, this study was approved by the institutional review board/ethics committee of CNUH (IRB No. CNUH-2015-250, Approved on 24 November 2021). All participants provided written informed consent. All procedures were conducted in compliance with the principles outlined in the Declaration of Helsinki with full respect to patient confidentiality and data protection standards. 

Data were extracted through a detailed review of the hospital’s electronic medical records. Demographic variables collected included age at diagnosis, sex, disease duration, and any coexisting medical conditions that could influence disease presentation or hematological parameters. Clinical data focused on both classification criteria manifestations and non-criteria features, with a particular emphasis on hematological abnormalities such as thrombocytopenia and hemolytic anemia. Other non-criteria manifestations reviewed included APS-related nephropathy and dermatologic findings, such as livedo reticularis. Laboratory findings at the time of diagnosis and during follow-up were comprehensively analyzed. These included the three major aPLs (lupus anticoagulant [LAC], anticardiolipin antibodies [aCL], and anti-β2 glycoprotein I antibodies), as well as additional serologic markers relevant in systemic autoimmune conditions, such as antinuclear antibodies (ANA), extractable nuclear antigens, and other autoantibodies. For LAC testing, we used dilute Russell viper venom time (dRVVT) with confirmatory and mixing studies (STA^®^-Staclot dRVVT, Stago) to establish positivity. The cut-off values for all aPL antibodies were defined as >99th percentile of normal based on the manufacturer’s instructions. Additionally, results of the direct antiglobulin test (Coombs test) were collected to identify patients with autoimmune hemolytic anemia. We defined clinically relevant AIHA as a positive Coombs test accompanied by at least one additional laboratory marker of hemolysis (e.g., elevated LDH, low haptoglobin, indirect hyperbilirubinemia, or reticulocytosis). Isolated Coombs positivity without hemolytic evidence was excluded from the AIHA group.

Furthermore, therapeutic data were obtained to assess the potential association between treatment and the presence or severity of hematological manifestations. The medications reviewed included anticoagulants such as warfarin and direct oral anticoagulants (DOACs), antiplatelet agents such as aspirin, and immunomodulatory therapies including hydroxychloroquine, corticosteroids, and other immunosuppressants, if applicable. Treatment regimens at diagnosis and any changes during follow-up were noted when available. For disease burden evaluation, we utilized the Damage Index for Antiphospholipid Syndrome (DIAPS), a validated instrument that quantifies cumulative organ damage in APS patients based on both clinical and laboratory features.

This comprehensive collection of clinical, serological, and treatment-related data provides a detailed view of the hematological manifestations of APS in a real-world clinical setting. The longitudinal nature of our dataset, spanning 10 years, enabled robust analysis of both prevalence patterns and serological associations with hematological abnormalities.

The longitudinal nature of our dataset, spanning 10 years, enabled robust analysis of both prevalence patterns and serological associations with hematological abnormalities.

### Statistical Analysis

All data processing and statistical analyses were conducted using the IBM SPSS statistics for windows version 21.0 (SPSS, Chicago, IL, USA). The selection of this statistical platform allowed for efficient data handling, descriptive summarization, and inferential testing, which is appropriate for retrospective clinical data. Prior to analysis, all collected data were systematically reviewed for completeness and accuracy, and any missing or ambiguous entries were clarified through a chart review, where possible. Outlier values were assessed for their plausibility within the clinical context and were retained if they reflected the true biological variability. Descriptive statistics were used to summarize the baseline characteristics of the study population. Continuous variables were primarily presented as medians ± standard deviation to convey central tendency and variability. Categorical variables are expressed as absolute frequencies and corresponding percentages. Univariate and multivariate logistic regression analyses were performed to evaluate the factors associated with the occurrence of hemolytic anemia in patients with APS. There models were designed to estimate the strength of the association between candidate predictor variables such as demographic data, serological profiles, and treatment exposures, and the outcome variable, which was defined as the presence or absence of Coombs-positive hemolytic anemia. In the univariate analysis, each potential factor was tested independently to assess its association with hemolytic anemia. Variables that demonstrated statistical significance or clinical relevance in the univariate analysis were subsequently included in the multivariate logistic regression model to adjust for potential confounders. Odds ratios (ORs) and corresponding 95% confidence intervals (CIs) were calculated to quantify the strength and precision of the observed associations. OR provides an estimate of the likelihood of hemolytic anemia occurring in association with a given variable after adjusting for other covariates in the model. Statistical significance was assessed using a two-sided *p*-value, with a threshold of *p* < 0.05, which was considered to indicate a result unlikely due to chance alone. All statistical tests were performed at the 95% CI. The analysis was conducted with careful consideration of potential multicollinearity among the independent variables, and the model fit was evaluated using standard goodness-of-fit measures. Sensitivity analyses were performed where appropriate to verify the robustness of our findings. Through this comprehensive statistical approach, we aimed to identify clinically meaningful predictors of hemolytic anemia within the broader context of APS.

## 3. Results

Overall, 346 patients who met the revised Sydney criteria for definite APS were included in the final analysis. Among the total 346 patients, 260 (75.1%) were classified as having primary APS, while 86 (24.9%) had secondary APS, primarily associated with systemic lupus erythematosus (SLE). Table 1 shows Baseline characteristics in total patients with APS. The mean age was 47.1 ± 13.1 years, indicating a predominance of middle-aged adults. A substantial majority of patients were female (71.7%), which is consistent with the known female predominance observed in systemic autoimmune diseases. The average disease duration for diagnosis to the time of data collection was 52.9 ± 8.2 months, reflecting a relatively chronic disease course across the study population. Most patients (93.3%) were covered by the Korean National Health Insurance system, suggesting relatively uniform access to healthcare services and treatment availability within the cohort. Regarding comorbidities, thromboembolic events were highly prevalent. Pulmonary thromboembolism was the most frequently observed vascular complication, affecting 33.5% of patients. Cerebrovascular accident, another major thrombotic manifestation, were present in 32.1% of cases. These findings highlight the high burden of vascular involvement in patients with APS. Notably, hypertension was the most common nonautoimmune chronic condition, reported in 29.1% of the cohort, and may have contributed to the overall thrombotic risk. Other notable comorbidities include systemic lupus erythematosus (SLE), deep vein thrombosis, ischemic heart disease, diabetes mellitus, and malignancy in 24.9%, 16.8%, 10.2%, 9.2%, and 1.2% of the cohort, respectively. The relatively high co-prevalence of SLE is consistent with the known overlap between primary and secondary APS and may have implications for the interpretations of immunological markers.

Table 2 presented clinical symptoms and laboratory findings in total patients with APS. Regarding non-criteria manifestations, hematological abnormalities were relatively common. Thrombocytopenia, defined as a platelet count below 130,000/μL, was observed in 34.5% of patients. A positive direct Coombs test result suggestive of autoimmune hemolytic anemia was identified in 16.5% of the cohort. These findings highlight the substantial burden of hemolytic involvement in APS, particularly considering that neither thrombocytopenia nor hemolytic anemia is currently included in the formal classification criteria. Other non-criteria features included nephropathy (8.4%), which may reflect either immune complex-mediated glomerular disease or chronic hypertension, and livedoid reticularis (9.8%), a characteristic cutaneous manifestation associated with vascular pathology in APS. Serological analysis revealed that LAC was the most frequently detected aPL (66.8% of patients). Anti-β2 glycoprotein I IgG was positive in 41.9% of patients, while anticardiolipin IgM and IgG antibodies were each detected in approximately 25% of cases (25.1% and 24.9%, respectively). Anti-β2 glycoprotein I IgM was present in 25.1% of patients. Notably, a significant proportion of the cohort (58.4%) tested positive for ANA, and 41.9% tested positive for anti-nucleosome antibodies. These findings suggest a strong autoimmune background in many patients, supporting the theory that APS exists on a spectrum similar to that of other systemic autoimmune diseases.

Table 3 shows the treatment history in total patients with APS. Treatment history shows that the most commonly used therapies were hydroxychloroquine (74.9%) and aspirin (66.5%). The frequent use of hydroxychloroquine reflects both its immunomodulatory effects and its common use in patients with coexisting SLE. Anticoagulant and anti-platelet therapies varied, though they included heparin (28.1%), DOACs (25.1%), clopidogrel (16.4%), warfarin (12.4%), respectively. Other agents included glucocorticoids (25.1%), tacrolimus (11.1%), and intravenous immunoglobulin (IVIG) (1.2%), which were typically reserved for refractory or severe immune-mediated hematologic manifestations. 

Table 4 presented univariate and multivariate logistic regression analyses of the factors with hemolytic anemia. In univariate logistic regression analysis aimed at identifying factors associated with hemolytic anemia, LAC positivity emerged as a strong predictor (OR 4.216, 95% CI: 2.326–7.640, *p* < 0.001). Other factors that demonstrated statistically significant associations were anticardiolipin IgG positivity (OR 7.170, *p* = 0.007), triple antibody positivity (OR 3.638, *p* = 0.002), and presence of diabetes mellitus (OR 2.084, *p* = 0.007). Interestingly, the DIAPS was inversely associated with hemolytic anemia (OR 0.547, *p* = 0.002), suggesting that patients with greater cumulative organ damage may be less likely to develop this specific hematological complication, potentially due to immune exhaustion or differences in the disease phenotype. However, in the multivariate logistic regression analysis, which was adjusted for confounding factors and provided a more robust evaluation of independent associations, only LAC positivity remained significantly associated with hemolytic (adjusted OR 3.557, 95% CI: 1.355–9.335, *p* = 0.003). Other variables that were significant in the univariate analysis, including triple antibody positivity and anticardiolipin IgG levels, were not statistically significant in the adjusted model. This suggests that LAC plays a distinct and potentially pathogenic role in the development of hemolytic anemia, independent of other serologic risk factors. 

## 4. Conclusions

In this retrospective cohort study involving a well-characterized population of patients with APS, we found that LAC positivity was independently associated with hemolytic anemia. Conversely, other aPL subtypes, including anticardiolipin and triple antibody positivity, did not demonstrate a statistically significant correlation with this hematologic manifestation. Although thrombocytopenia was observed at a relatively high frequency within the study population, it did not show a meaningful association with specific serological profiles in the multivariate regression analysis. These findings suggest that LAC plays a distinct and potentially pathogenic role in the development of autoimmune hemolytic anemia in patients with APS. This highlights an important serological-clinical relationship that has been under-recognized in the current literature and may have significant implications for patient monitoring and management. 

Historically, research on APS has primarily focused on thrombotic and obstetric complications, which form the basis of established classification criteria. Triple positivity, defined as the concurrent presence of LAC, anti-cardiolipin antibody, and anti-beta2 glycoprotein antibodies, has been consistently identified as a marker of high thrombotic risk and is often used to guide clinical decision-making in terms of anticoagulation strategies and long-term management [5,6,10]. However, this emphasis on vascular complications has inadvertently led to the relative neglect of non-criteria manifestations, such as hematologic abnormalities, neuropsychiatric symptoms, and cutaneous signs, which are frequently encountered in real-world clinical practice but have not yet been incorporated into diagnostic algorithms. Among the hematologic features, thrombocytopenia and hemolytic anemia are two of the most frequently reported manifestations; however, they remain poorly defined in terms of their immunologic correlates and clinical significance. Previous studies have suggested a potential link between aPL and cytopenia; however, the strength and specificity of these associations have varied considerably across studies, partly because of differences in study design, population size, and definitions of hematologic outcomes [11,12]. Galli et al. previously demonstrated that LAC is a stronger predictor of thrombotic events than anti-cardiolipin antibodies, raising the possibility that LAC may have broader pathogenic implications beyond thrombosis [7]. However, the role of LAC in non-thrombotic complications, particularly hematological abnormalities, remains largely speculative. 

Our study contributes novel data in this area by demonstrating a statistically significant and independent association between LAC positivity and hemolytic anemia, even after adjusting for potential confounders. Several previous studies have explored the relationship between LAC and hematologic abnormalities. For instance, Devreese et al. [13] and Arnaud et al. [14] both reported a higher frequency of hemolytic anemia in LAC-positive patients, although these findings were primarily descriptive. In contrast, studies by Oku et al. [15] and Miyakis et al. did not find a consistent association, possibly due to smaller sample sizes or differing definitions of AIHA. Moreover, a large multicenter cohort by Belizna et al. identified LAC as a predictor not only of thrombosis but also of non-criteria manifestations such as thrombocytopenia and anemia, supporting the idea that LAC contributes broadly to APS pathogenesis. Our findings align with these reports and extend them by demonstrating an independent association in a multivariate framework. This suggests that a more targeted immunopathogenic mechanism links LAC to red cell destruction, possibly through complement activation or cross-reactivity with erythrocyte antigens. These results support the hypothesis that specific aPL subtypes contribute differentially to the phenotypic expression of APS, thereby challenging the conventional model that emphasizes triple positivity as the primary serological risk stratifier. Conversely, our findings suggest that individual aPL components, such as LAC, may independently drive certain clinical features, particularly those outside the scope of the current classification criteria. 

One of the major strengths of this study was the use of a relatively large and diverse patient cohort along with a longitudinal dataset spanning a full decade of clinical observation. This extended timeframe enabled us to capture a broad array of clinical presentations and temporal patterns in a real-world population of patients with APS. Unlike many previous investigations that were limited by small sample sizes, restrictive inclusion criteria, or a narrow focus on thrombotic outcomes, this study deliberately emphasized non-criteria manifestations, especially hematological abnormalities, which remain an unmet need in both clinical recognition and research exploration. By identifying a specific serological–hematologic link in this context, this study deepens the current understanding of APS pathogenesis and offers important implications for future revisions of the diagnostic and classification frameworks. For instance, if corroborated in larger prospective multicenter studies, hematologic manifestations such as hemolytic anemia may warrant formal inclusion as secondary or supportive criteria in the classification of APS, particularly in patients.

Furthermore, the clinical implications of these findings extend to patient monitoring and personalized care. Patients who demonstrate isolated or predominant LAC positivity may benefit from more vigilant surveillance for hematological complications, even in the absence of classic APS features. These findings emphasize the importance of maintaining a high level of clinical suspicion for immune-mediated cytopenia in LAC-positive individuals. Moreover, these results may inform therapeutic decision-making regarding the use of immunosuppressive agents, adjunctive therapies, or the frequency of hematological monitoring. Ultimately, this study contributes to the growing recognition that APS is not a uniform disease entity but rather a clinically heterogeneous syndrome with diverse immunologic drivers. Personalized risk stratification may be enhanced by integrating serological markers and non-criteria clinical features into clinical algorithms.

Despite its strengths, this study has several limitations that warrant consideration. First, the retrospective study design precludes establishing causality and introduces potential biases that may affect data integrity. Second, dataset lacked certain laboratory parameters, such as the Global APS score (GAPSS) or adjusted GAPSS, which could have provided additional mechanistic insights into the immunologic landscape of patients with hematologic involvement. We acknowledge that GAPSS incorporates both laboratory (e.g., aCL, anti-β2GPI, LAC, aPS/PT) and non-laboratory clinical parameters, including hypertension and dyslipidemia. The absence of GAPSS in our dataset therefore limits assessment of the broader immuno-clinical profile associated with hematologic manifestations. The absence of these parameters restricts the ability to explore the interplay between immune activation and hematological abnormalities. Third, the study was conducted at a single center, which may limit generalizability to broader populations. And although our study utilized the revised Sydney criteria to define APS, we were unable to fully apply the 2023 ACR/EULAR classification system due to the retrospective nature of the dataset and the absence of certain required clinical domains (e.g., echocardiographic data for heart valve disease or detailed microvascular assessments). This limits direct comparison with newer classification models and underscores the need for prospective data collection aligned with evolving criteria.

In conclusion, this study identifies LAC positivity as an independent serological risk factor for hemolytic anemia in patients with APS. These results underscore the need for heightened awareness and clinical attention to non-criteria hematological manifestations, which may serve as early indicators of immune dysregulation and contribute to the broader clinical phenotype of APS. By uncovering a specific and reproducible association between LAC and hemolytic anemia, this study supports an expanded framework for evaluating patients with APS extending beyond the traditional thrombotic and obstetric domains. These findings reinforce the importance of distinguishing between different aPL profiles when evaluating hematological manifestations in APS and provide further support for the central role of LAC in the broader clinical expression of the syndrome. Continued research is warranted to further elucidate the mechanistic pathways linking LAC to red cel destruction and to validate these findings across larger and more diverse patient cohorts. Such efforts may ultimately inform future revisions of the APS classification criteria and enable more tailored evidence-based approaches to the diagnosis, monitoring, and treatment of patients with complex APS phenotypes.

## Figures and Tables

**Table 1 medicina-61-01364-t001:** Baseline characteristics in total patients with antiphospholipid syndrome.

Variables	
Age at enrollment	47.1 ± 13.1
Women (%)	71.7
Disease duration (months)	52.9 ± 8.2
Health insurance (%)	93.3
**Comorbidities (%)**	
Hypertension	29.1
Diabetes	9.2
Hyperlipidemia Ischemic heart disease Cerebrovascular accident	32.1 10.2 24.9
Deep vein thrombosis	16.8
Pulmonary thromboembolism	33.5
Systemic lupus erythematosus	24.9
Malignancy	1.2
**Thrombotic Events (Sydney criteria)**	
Arterial thrombosis (%)	27.1
–Stroke or TIA	13.1
–Myocardial infarction	8.5
–Other arterial thrombosis	5.5
Venous thrombosis (%)	45.2
–Deep vein thrombosis	19.1
–Pulmonary thromboembolism	22.1
–Other venous thrombosis	4.0
Small-vessel thrombosis (%)	1.7
**Pregnancy Morbidity (Sydney criteria)**	
≥1 Fetal death after 10 weeks (%)	15.2
≥1 Premature birth before 34 weeks (%)	9.3
≥3 Unexplained consecutive abortions (%)	1.5

Data were presented as mean ± standard deviation for continuous variables. TIA: Transient Ischemic Attack.

**Table 2 medicina-61-01364-t002:** Clinical symptoms and Laboratory findings in total patients with antiphospholipid syndrome.

Variables (%)	
Thrombocytopenia	34.5
Direct coombs test	16.5
Nephropathy	8.4
Livedoid reticularis	9.8
Lupus anticoagulant	66.8
Anti-cardiolipin IgG	24.9
Anti-cardiolipin IgM	25.1
Anti-beta 2 glycoprotein IgG	41.9
Anti-beta 2 glycoprotein IgM Triple positivity	25.1 11.1
Antinuclear antibody	58.4
Anti-double strand DNA	16.5
Anti-Smith	3.5
Anti-RNP	7.7
Anti-Ro/SS-A	24.9
Anti-La/SS-B	8.4
Anti-centromere	1.2
Anti-Scl70	0.8
Anti-histone	16.8
Anti-ribosomal P	2.5
Anti-nucleosome	41.9

Data were presented as mean ± standard deviation for continuous variables.

**Table 3 medicina-61-01364-t003:** Treatment history in total patients with antiphospholipid syndrome.

Variables (%)	
Aspirin	66.5
Clopidogrel	16.4
Heparin	28.1
Warfarin	12.4
DOAC	25.1
Hydroxychloroquine	74.9
Glucocorticoids	25.1
Tacrolimus	11.1
IVIG	1.2

DOAC: direct anticoagulant, IVIG: intravenous immunoglobulin.

**Table 4 medicina-61-01364-t004:** Univariate and multivariate logistic regression analyses of the factors associated with hemolytic anemia.

Variable	Univariate Analysis		Multivariate Analysis	
	Odds Ratio (95% CI)	*p*-Value	Odds Ratio (95% CI)	*p*-Value
Age	0.994 (0.976–1.013)	0.547		
Women	1.087 (0.628–1.881)	0.766		
Disease duration	0.443 (0.168–1.167)	0.099		
SLE	0.981 (0.918-1.049)	0.581		
Lupus anticoagulant	4.216 (2.326–7.640)	<0.001	3.557 (1.355–9.335)	0.003
Anticardiolipin IgG	7.170 (1.735–29.638)	0.007	1.449 (0.199–10.563)	0.714
Triple positivity	3.638 (1.611–8.215)	0.002	1.259 (0.654–2.421)	0.491
DIAPS	0.547 (0.402–0.746)	0.002	0.831 (0.614–1.126)	0.232

SLE: Systemic Lupus Erythematosus, DIAPS: Disease activity index for antiphospholipid syndrome.

## Data Availability

The original contributions presented in this study are included in the article/Appendix A. Further inquiries can be directed to the corresponding author(s).

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
