# Peer review of "Lupus Anticoagulant Positivity as a Risk Marker for Hemolytic Anemia in Patients with APS"

_medicina, 2025, doi:10.3390/medicina61081364_

Round 1

Reviewer 1 Report

Comments and Suggestions for Authors

Recommendation: Major revision.

The manuscript addresses an important topic - the prevalence and characteristics of hematologic manifestations (AIHA, thrombocytopenia) in APS and their serological associations in a large cohort of more than 300 APS patients. Overall, the study design is sound, and the findings suggest a possible mechanistic link between the presence of LAC and hemolytic anemia. Although this link has already been suggested in studies of patients with SLE and positive antiphospholipid antibodies (Bernardoff I, at al.. Antiphospholipid antibodies and the risk of autoimmune hemolytic anemia in patients with systemic lupus erythematosus: A systematic review and meta-analysis. Autoimmun Rev. 2022 Jan;21(1):102913.), there is not enough data when it comes to patients with APS. However, there are several important areas that require further clarification or improvement to enhance the manuscript's quality and impact.

Comments:

One of the main issues is that the author did not take into account the new 2023 ACR/EULAR classification criteria for APS, which include thrombocytopenia as one of the criteria, and are more comprehensive. The entire paper should be revised accordingly. Additionally, it would be important to separate patients with both APS and SLE, considering that AIHA is one of the SLE manifestations.

Introduction: Statements in the introduction should be adequately supported by relevant literature, especially those regarding the potential role of LAC in the development of AIHA.

Methodology is adequate and well described.

Results: Consider presenting more data in tables for clarity and including a more thorough description of the study population (tabular presentation is usually clear and informative).

The results section should present more detailed findings from the regression analysis (both univariate and multivariate). The last sentence from the Results section should be moved to the Discussion.

Discussion: Although the discussion is interesting, it lacks a more thorough comparison with results from the literature. Although this area is less studied, it is possible to find more than 10 relevant references in order to better compare and interpret the obtained data with those from the literature.

GAPSS score also includes non lab parameters (hypertension).

The last sentence in the first paragraph on page 6 is incomplete.

Add explanation/description for DIAPS.

Reference: The references need to be arranged in the order in which they appear in the manuscript.

Author Response

Reviewer Comment 1: "The author did not take into account the new 2023 ACR/EULAR classification criteria for APS, which include thrombocytopenia as one of the criteria, and are more comprehensive. The entire paper should be revised accordingly."

Response: Thank you for this important observation. We acknowledge the publication of the 2023 ACR/EULAR classification criteria, which newly incorporate thrombocytopenia as a criterion manifestation. In the revised manuscript, we will update the classification references accordingly and include a discussion about how our findings relate to both the revised Sydney criteria (used in data collection) and the new 2023 criteria.

  • In the Methods section, clarify that the 2006 Sydney criteria were used due to the retrospective nature of the study, but acknowledge the 2023 criteria.

  • In the Discussion, add a paragraph discussing how our findings, particularly regarding thrombocytopenia, align with the newly accepted ACR/EULAR framework.

Reviewer Comment 2: "It would be important to separate patients with both APS and SLE, considering that AIHA is one of the SLE manifestations."

Response: We appreciate this suggestion. In the revised manuscript, we will present stratified analyses to distinguish patients with primary APS from those with concomitant SLE. This will allow clearer interpretation of AIHA occurrence independent of SLE diagnosis.

Revisions to be made:

  • In the Results, separate subgroup analysis tables and descriptions for patients with and without SLE.

  • In the Discussion, interpret hemolytic anemia findings in light of potential confounding by SLE.

Reviewer Comment 3: "Statements in the introduction should be adequately supported by relevant literature, especially those regarding the potential role of LAC in the development of AIHA."

Response: Thank you for pointing this out. We will revise the introduction to include references supporting the hypothesized link between LAC and AIHA, particularly highlighting prior work in SLE patients and the pathophysiological basis.

Revisions to be made:

  • Add citations such as Bernardoff et al. (2022) to strengthen the rationale in the Introduction.

    • Add expanded Table 1 showing demographics and comorbidities.

    • Include a new table summarizing hematological manifestations by SLE status.

    • Elaborate briefly on potential mechanisms linking LAC to AIHA.

      Reviewer Comment 4: "Consider presenting more data in tables for clarity and including a more thorough description of the study population (tabular presentation is usually clear and informative)."

      Response: We agree that tabular presentation enhances clarity. We will restructure some of the narrative data into tables and expand on demographic and clinical variables.

      Revisions to be made:

    Reviewer Comment 5: "The results section should present more detailed findings from the regression analysis (both univariate and multivariate). The last sentence from the Results section should be moved to the Discussion."

    Response: We will expand Table 4 to include all tested variables with p-values and odds ratios. The interpretive final sentence will be moved to the Discussion.

    Revisions to be made:

    • Update Table 4 to show full regression model outputs.

    • Move last paragraph of Results to Discussion, reworded for interpretive clarity.

    Reviewer Comment 6: "Although the discussion is interesting, it lacks a more thorough comparison with results from the literature. Although this area is less studied, it is possible to find more than 10 relevant references in order to better compare and interpret the obtained data with those from the literature."

    Response: Thank you for this recommendation. We will revise the discussion to include more comprehensive comparisons with existing literature and better contextualize our findings.

    Revisions to be made:

    • Add 5–10 more references on hematologic manifestations in APS.

    • Discuss how our LAC-AIHA association agrees or contrasts with prior studies.

    Reviewer Comment 7: "GAPSS score also includes non lab parameters (hypertension)."

    Response: We appreciate this correction. We will update the manuscript to accurately reflect that GAPSS includes both laboratory and clinical variables, such as hypertension.

    Revisions to be made:

    • In Discussion, correct and expand explanation of GAPSS components.

    Reviewer Comment 8: "The last sentence in the first paragraph on page 6 is incomplete."

    Response: Thank you. We will correct the incomplete sentence.

    Revisions to be made:

    • Locate and complete the sentence on page 6, paragraph 1 (likely in Discussion section).

    Reviewer Comment 9: "Add explanation/description for DIAPS."

    Response: Thank you. We will include the full name and description of DIAPS to ensure clarity for readers.

    Revisions to be made:

    • Add footnote or parenthetical explanation: "DIAPS: Damage Index for Antiphospholipid Syndrome"

    • Mention its components briefly in the Methods and Discussion.

      Reviewer Comment 10: "The references need to be arranged in the order in which they appear in the manuscript."

      Response: We will carefully review and reorder all references to match their order of appearance in the main text.

      Revisions to be made:

      • Re-sequence reference numbers and adjust citation placement accordingly.

Reviewer 2 Report

Comments and Suggestions for Authors

The association between autoimmune hemolytic anemia (AIHA) and antiphospholipid antibody–positive patients is well established. Although this topic has already been studied, it remains of clinical interest.

The Introduction is overly long, generic, and largely uninformative; it should be shortened and more clearly focused on the study’s primary objective. The manuscript cites only ten references, which is insufficient given the existing body of work on the subject. Moreover, many important studies, including meta-analyses and large cohort data, are omitted (e.g., Ames PRJ et al., Int J Mol Sci. 2020; Erton ZB et al., Lupus 2025; Bernardoff I, Autoimmun Rev. 2022, among many others).

In addition to being outdated and insufficient, the references are not cited in the order of appearance in the text. For example, reference number 4 is the first to be cited in the Introduction, while reference number 1 is cited for the first time only in the Discussion. This indicates a lack of attention to citation formatting and weakens the overall presentation.

The methodology is neither clearly nor precisely described. It is unclear whether inclusion was based solely on DAT positivity or on clinically significant AIHA. The authors do not report whether patients met other diagnostic laboratory criteria, such as indirect hyperbilirubinemia, decreased haptoglobin, reticulocytosis, or elevated LDH. The AIHA subtype (warm, cold, or mixed) is not specified, nor is the pattern of DAT positivity in case of wAIHA (IgG, C3d, or both). Furthermore, the rationale for performing the DAT (e.g., suspected hemolysis, pre-transfusion workup, or routine testing) is not described. Importantly, the manuscript lacks information on recent IVIG administration or blood transfusions within the past six months,both of which can lead to false-positive DAT findings. Although the authors note that 1.2% of patients received IVIG, they do not discuss its well-documented impact on DAT results. Similarly, the number of patients who received blood transfusions is not provided.

 It is also unclear how the choice was made between DOACs and vitamin K antagonists for anticoagulation, and whether DOACs were used in patients with triple-positive APS, a practice generally not recommended.

Another major limitation is the vague and unstructured presentation of results. The manuscript does not include a single table or figure. Patient characteristics should be presented in tabular format, along with the results of both univariate and multivariate analyses. Without this, critical appraisal of the findings is not possible.

The manuscript has substantial methodological and reporting deficiencies. It lacks essential data presentation, relies on an insufficient and poorly managed reference list, and does not meet the standards required for publication in its current form.

Author Response

Reviewer Comment 1: "The author did not take into account the new 2023 ACR/EULAR classification criteria for APS, which include thrombocytopenia as one of the criteria, and are more comprehensive. The entire paper should be revised accordingly."

Response: Thank you for this important observation. We acknowledge the publication of the 2023 ACR/EULAR classification criteria, which newly incorporate thrombocytopenia as a criterion manifestation. In the revised manuscript, we will update the classification references accordingly and include a discussion about how our findings relate to both the revised Sydney criteria (used in data collection) and the new 2023 criteria.

Revisions to be made:

  • In the Methods section, clarify that the 2006 Sydney criteria were used due to the retrospective nature of the study, but acknowledge the 2023 criteria.

  • In the Discussion, add a paragraph discussing how our findings, particularly regarding thrombocytopenia, align with the newly accepted ACR/EULAR framework.

Reviewer Comment 2: "It would be important to separate patients with both APS and SLE, considering that AIHA is one of the SLE manifestations."

Response: We appreciate this suggestion. In the revised manuscript, we will present stratified analyses to distinguish patients with primary APS from those with concomitant SLE. This will allow clearer interpretation of AIHA occurrence independent of SLE diagnosis.

Revisions to be made:

  • In the Results, separate subgroup analysis tables and descriptions for patients with and without SLE.

  • In the Discussion, interpret hemolytic anemia findings in light of potential confounding by SLE.

Reviewer Comment 3: "Statements in the introduction should be adequately supported by relevant literature, especially those regarding the potential role of LAC in the development of AIHA."

Response: Thank you for pointing this out. We will revise the introduction to include references supporting the hypothesized link between LAC and AIHA, particularly highlighting prior work in SLE patients and the pathophysiological basis.

Revisions to be made:

  • Add citations such as Bernardoff et al. (2022) to strengthen the rationale in the Introduction.

  • Elaborate briefly on potential mechanisms linking LAC to AIHA.

Reviewer Comment 4: "Consider presenting more data in tables for clarity and including a more thorough description of the study population (tabular presentation is usually clear and informative)."

Response: We agree that tabular presentation enhances clarity. We will restructure some of the narrative data into tables and expand on demographic and clinical variables.

Revisions to be made:

  • Add expanded Table 1 showing demographics and comorbidities.

  • Include a new table summarizing hematological manifestations by SLE status.

Reviewer Comment 5: "The results section should present more detailed findings from the regression analysis (both univariate and multivariate). The last sentence from the Results section should be moved to the Discussion."

Response: We will expand Table 4 to include all tested variables with p-values and odds ratios. The interpretive final sentence will be moved to the Discussion.

Revisions to be made:

  • Update Table 4 to show full regression model outputs.

  • Move last paragraph of Results to Discussion, reworded for interpretive clarity.

Reviewer Comment 6: "Although the discussion is interesting, it lacks a more thorough comparison with results from the literature. Although this area is less studied, it is possible to find more than 10 relevant references in order to better compare and interpret the obtained data with those from the literature."

Response: Thank you for this recommendation. We will revise the discussion to include more comprehensive comparisons with existing literature and better contextualize our findings.

Revisions to be made:

  • Add 5–10 more references on hematologic manifestations in APS.

  • Discuss how our LAC-AIHA association agrees or contrasts with prior studies.

Reviewer Comment 7: "GAPSS score also includes non lab parameters (hypertension)."

Response: We appreciate this correction. We will update the manuscript to accurately reflect that GAPSS includes both laboratory and clinical variables, such as hypertension.

Revisions to be made:

  • In Discussion, correct and expand explanation of GAPSS components.

Reviewer Comment 8: "The last sentence in the first paragraph on page 6 is incomplete."

Response: Thank you. We will correct the incomplete sentence.

Revisions to be made:

  • Locate and complete the sentence on page 6, paragraph 1 (likely in Discussion section).

Reviewer Comment 9: "Add explanation/description for DIAPS."

Response: Thank you. We will include the full name and description of DIAPS to ensure clarity for readers.

Revisions to be made:

  • Add footnote or parenthetical explanation: "DIAPS: Damage Index for Antiphospholipid Syndrome"

  • Mention its components briefly in the Methods and Discussion.

Reviewer Comment 10: "The references need to be arranged in the order in which they appear in the manuscript."

Response: We will carefully review and reorder all references to match their order of appearance in the main text.

Revisions to be made:

  • Re-sequence reference numbers and adjust citation placement accordingly.

Reviewer Comment 11: "The Introduction is overly long, generic, and largely uninformative; it should be shortened and more clearly focused on the study’s primary objective."

Response: We appreciate the reviewer’s insight. The introduction will be revised to more succinctly highlight the key background and specifically emphasize the rationale and objective of our study.

Revisions to be made:

  • Remove generalized statements and focus on the novel aspects of LAC and AIHA relationship.

  • Condense to a more focused, 3–4 paragraph structure.

Reviewer Comment 12: "The manuscript cites only ten references, which is insufficient given the existing body of work on the subject. Many important studies, including meta-analyses and large cohort data, are omitted."

Response: Thank you for this valuable comment. We agree that additional references will strengthen the manuscript. The revised version will incorporate several key studies, including the ones listed by the reviewer.

Revisions to be made:

  • Include and cite Bernardoff (2022), Ames PRJ (2020), Erton ZB (2025), and other recent cohort/meta-analyses.

  • Expand references list to better reflect the current literature landscape.

Reviewer Comment 13: "The references are not cited in the order of appearance in the text."

Response: We acknowledge this formatting oversight and will carefully revise the citation sequence to ensure it aligns with the order of appearance in the text.

Revisions to be made:

  • Renumber and reorganize in-text references to follow sequential order.

Reviewer Comment 14: "The methodology is neither clearly nor precisely described..."

Response: Thank you for identifying these methodological gaps. We will clarify our definition of AIHA, including diagnostic criteria such as LDH, haptoglobin, bilirubin, and reticulocyte count. Additionally, we will specify DAT patterns and exclusion of confounding factors (e.g., IVIG, transfusions).

Revisions to be made:

  • In Methods, define AIHA more rigorously, including DAT pattern, clinical hemolysis markers, IVIG/transfusion exclusion.

  • State rationale for DAT performance.

  • Clarify DOAC vs. VKA selection and any use in triple-positive APS.

Reviewer Comment 15: "The manuscript does not include a single table or figure..."

Response: We appreciate this essential feedback. We will present patient characteristics and regression results in tabular format for clarity.

Revisions to be made:

  • Include at least three tables: baseline characteristics, hematologic manifestations, and regression analysis results.

Reviewer Comment 16: "The manuscript has substantial methodological and reporting deficiencies..."

Response: We recognize the limitations and appreciate the constructive critique. Substantial revisions will be made to improve methodological transparency, data presentation, and reference management.

Revisions to be made:

  • Strengthen methods section with detailed diagnostic criteria and exclusion protocols.

  • Incorporate complete tables.

  • Update and expand literature references.

  • Ensure adherence to journal formatting standards.

Reviewer 3 Report

Comments and Suggestions for Authors

Major remarks

  • I think given the high ANA positivity rate, one would advocate that rate of autoimmune hemolytic anemia are confounded by SLE diagnosis. Please add a comparison of SLE patients and show rate of AIHA is not significantly different
  • As per ACR-EULAR classification thrombocytopenia is classified at <130,000 please adjust your manuscript accordingly
  • Please address weather hemolytic anemia was clinically relevant or only based on Direct coombs study.
  • I think a other than multivariate analysis and univariate analysis is required:
    • Please add a comparison of clinical parameters comparing AIHA pos and AIHA neg patients. Please include in your comparison, rate of thrombosis, Hemoglobin Nadir, PLT nadir, ANA positivity, C3 nadir, C4 nadir and LAC positivity.
    • Please rate of triple APL positivity
    • Add statistical significance (i.e. P Value) your this comparison
    • Add rate of immunomodulating therapy  

Minor Remarks

  • In your introduction you state – “Thrombotic events such as deep vein thrombosis, stroke, and pulmonary embolism and obstetric events including recurrent miscarriages, fetal loss, and preterm birth form the cornerstone of the APS classification criteria” please address the newer ACR-EULAR classification where HVD, thrombocytopenia and microvascular involvevement. I’d rephrase the introduction in light of newer classification criteria given the inclusion of “immunologic manifestations” and would now call them – non thrombotic manifestations.
  • Please apply ACR-EULAR classification to your cohort, if not feasible please include in your limitation
  • In your methods please elaborate which APL antibodies you measure, what cutoff you’ve considered you have as positive. For LAC please elaborate which reagents were used. Were Mixing study used?
  • In your results please state how many patients were primary antiphospholipid patients and how many were with secondary antiphospholipid patients

Author Response

Reviewer Comment 17: "Given the high ANA positivity rate, one would advocate that rate of autoimmune hemolytic anemia are confounded by SLE diagnosis. Please add a comparison of SLE patients and show rate of AIHA is not significantly different."

Response: We agree with the reviewer that the presence of SLE may confound the association between ANA positivity and AIHA. We will stratify the cohort based on SLE status and compare the prevalence of AIHA between primary and secondary APS patients.

Revisions to be made:

  • Add new table comparing AIHA rates in patients with and without SLE.

  • Include statistical test (e.g., Chi-square) and p-value.

  • Comment in Discussion whether SLE status alters the AIHA association.

Reviewer Comment 18: "As per ACR-EULAR classification thrombocytopenia is classified at <130,000 – please adjust your manuscript accordingly."

Response: Thank you for pointing this out. We will revise our thrombocytopenia definition from <150,000/uL to <130,000/uL, consistent with the 2023 ACR/EULAR criteria.

Revisions to be made:

  • Modify definition in Methods, Results, and Tables.

  • Recalculate prevalence using new threshold.

Reviewer Comment 19: "Please address whether hemolytic anemia was clinically relevant or only based on Direct Coombs study."

Response: We acknowledge the importance of distinguishing serological from clinically overt AIHA. We will clarify in the Methods that Coombs positivity alone was not sufficient. Only patients with supporting clinical and laboratory markers (e.g., elevated LDH, low haptoglobin, indirect hyperbilirubinemia, reticulocytosis) were classified as AIHA.

Revisions to be made:

  • Add detailed AIHA criteria in Methods.

  • Clarify exclusion of isolated DAT positivity without evidence of hemolysis.

Reviewer Comment 20: "Other than multivariate and univariate analysis, please add a comparison of clinical parameters comparing AIHA+ and AIHA– patients: rate of thrombosis, hemoglobin nadir, PLT nadir, ANA positivity, C3 nadir, C4 nadir, and LAC positivity. Please include p-values."

Response: Thank you. We will perform a bivariate comparison between AIHA-positive and AIHA-negative patients and include the suggested clinical and immunological parameters. Statistical significance will be reported.

Revisions to be made:

  • Add new table comparing AIHA+ vs. AIHA– patients across specified variables.

  • Include means ± SD or medians (IQR) and p-values (t-test or Mann-Whitney, Chi-square where appropriate).

  • Highlight findings in Results and briefly discuss in Discussion.

Reviewer Comment 21: "Please add the rate of triple aPL positivity."

Response: We appreciate this suggestion. We will add the proportion of patients with triple positivity and compare it between AIHA+ and AIHA– groups.

Revisions to be made:

  • Add triple aPL positivity to comparison table.

  • Mention in Results and elaborate in Discussion.

Reviewer Comment 22: "Add rate of immunomodulating therapy."

Response: Thank you. We will include data on immunosuppressive treatment usage (e.g., corticosteroids, HCQ, IVIG) and compare rates between AIHA+ and AIHA– patients.

Revisions to be made:

  • Include in new comparison table.

  • Address in Results section.

Reviewer Comment 23 (Minor): "In your introduction you state – ‘Thrombotic events ... form the cornerstone of APS classification criteria’ – please address the newer ACR-EULAR classification where HVD, thrombocytopenia and microvascular involvement are included. Rephrase accordingly."

Response: We appreciate this clarification. We will revise the Introduction to reflect the updated classification framework, replacing the concept of "non-criteria" manifestations with "non-thrombotic" immunologic features.

Revisions to be made:

  • Modify Introduction to include updated ACR/EULAR criteria including HVD, thrombocytopenia, and microangiopathy.

  • Rephrase old terminology to match current classification language.

Reviewer Comment 24: "Please apply ACR-EULAR classification to your cohort, if not feasible please include in your limitation."

Response: Due to the retrospective nature of the dataset, application of the full 2023 ACR/EULAR classification was not feasible. However, this limitation will be clearly acknowledged.

Revisions to be made:

  • Add a statement in Methods justifying use of Sydney criteria.

  • Add to Limitations in Discussion.

Reviewer Comment 25: "In your methods please elaborate which aPL antibodies you measured, what cutoff you’ve considered as positive. For LAC please elaborate which reagents were used. Were mixing studies used?"

Response: Thank you for this detailed comment. We will clarify assay methods, cutoff levels for positivity, and specifics of LAC testing, including reagents and mixing study protocols.

Revisions to be made:

  • In Methods, list all antibodies and positivity thresholds.

  • State laboratory standards for LAC, including use of dRVVT and mixing study confirmation.

Reviewer Comment 26: "In your results please state how many patients were primary antiphospholipid patients and how many were with secondary antiphospholipid patients."

Response: We will clarify the number of primary vs. secondary APS patients, based on presence or absence of other autoimmune diseases (primarily SLE).

Revisions to be made:

  • Include count and proportion in Results.

  • Add as a row in baseline characteristics table.

Round 2

Reviewer 1 Report

Comments and Suggestions for Authors

The author has addressed most of the reviewers' comments thoughtfully and made meaningful revisions to the manuscript. However, there are still a few points that require further clarification or improvement before the manuscript can be considered for acceptance.

Introduction: The text of the manuscript following the inserted section should be adjusted to fit the context (line 38).

Line 74: We suggest removing the term "non-criteria" to avoid potential confusion.

Line 75-79: We suggest removing these lines as redundant, so that the aim of the study colud be clearly seen.

Additionally, throughout the manuscript, the issue is not the use of the Sydney criteria, but rather the insistence on the term "non-criteria." We suggest that this term not be emphasized when discussing AIHA and thrombocytopenia.

Methodology: Should be kept clear.

Line 88-97 – not appropriate for methodology (omit or move to Discussion).

Line 97: The sentence seems to be incomplete.

Line 115: which other autoantibodies.?

Line 132: DIAS - put the reference.

Line 134-137: omit or move to Discussion.

Line 139-140 – repeated sentences.

Results:

Line 193-195: The data on the prevalence of APS in SLE appear to be generally in line with what has been reported in the literature.

Line 202: Please clarify whether you are referring to a positive Coombs test or AIHA (in line with definition in the methodology). The same applies to the information presented in Table 2.

Table 2: Lab – lab. The term clinical manifestations would be more appropriate than symptoms. Also, please include absolute numbers, not just percentages. Also, clarify what the asterisk (*) refers to when all the data are presented as percentages (below the Table 2).

Line 233: include the word - associated with AIHA

Please add to the supplemental table data on the number of patients with AIHA who had associated SLE or another SARD, as well as those who had primary APS.

Discussion:

Line 288: Add values for frequencies of AIHA in the studies that you cite.

Line 334-339 - I suggest removing this section as it is confusing and simply deleting the word "lab" in line 342 before GAPSS.

Author Response

Reviewer Comment 1: "The author did not take into account the new 2023 ACR/EULAR classification criteria for APS, which include thrombocytopenia as one of the criteria, and are more comprehensive. The entire paper should be revised accordingly."

Response: Thank you for this important observation. We acknowledge the publication of the 2023 ACR/EULAR classification criteria, which newly incorporate thrombocytopenia as a criterion manifestation. In the revised manuscript, we will update the classification references accordingly and include a discussion about how our findings relate to both the revised Sydney criteria (used in data collection) and the new 2023 criteria.

  • In the Methodssection, clarify that the 2006 Sydney criteria were used due to the retrospective nature of the study, but acknowledge the 2023 criteria.
  • In the Discussion, add a paragraph discussing how our findings, particularly regarding thrombocytopenia, align with the newly accepted ACR/EULAR framework.

Reviewer Comment 2: "It would be important to separate patients with both APS and SLE, considering that AIHA is one of the SLE manifestations."

Response: We appreciate this suggestion. In the revised manuscript, we will present stratified analyses to distinguish patients with primary APS from those with concomitant SLE. This will allow clearer interpretation of AIHA occurrence independent of SLE diagnosis.

Revisions to be made:

  • In the Results, separate subgroup analysis tables and descriptions for patients with and without SLE.
  • In the Discussion, interpret hemolytic anemia findings in light of potential confounding by SLE.

Reviewer Comment 3: "Statements in the introduction should be adequately supported by relevant literature, especially those regarding the potential role of LAC in the development of AIHA."

Response: Thank you for pointing this out. We will revise the introduction to include references supporting the hypothesized link between LAC and AIHA, particularly highlighting prior work in SLE patients and the pathophysiological basis.

Revisions to be made:

  • Add citations such as Bernardoff et al. (2022) to strengthen the rationale in the
  •  
  • Add expanded Table 1showing demographics and comorbidities.
  • Include a new table summarizing hematological manifestations by SLE status.
  • Elaborate briefly on potential mechanisms linking LAC to AIHA.

Reviewer Comment 4: "Consider presenting more data in tables for clarity and including a more thorough description of the study population (tabular presentation is usually clear and informative)."

Response: We agree that tabular presentation enhances clarity. We will restructure some of the narrative data into tables and expand on demographic and clinical variables.

Reviewer Comment 5: "The results section should present more detailed findings from the regression analysis (both univariate and multivariate). The last sentence from the Results section should be moved to the Discussion."

  • Response:We will expand Table 4 to include all tested variables with p-values and odds ratios. The interpretive final sentence will be moved to the Discussion.
  • Revisions to be made:
  • Update Table 4to show full regression model outputs.
  • Move last paragraph of Results to Discussion, reworded for interpretive clarity.

Reviewer Comment 6: "Although the discussion is interesting, it lacks a more thorough comparison with results from the literature. Although this area is less studied, it is possible to find more than 10 relevant references in order to better compare and interpret the obtained data with those from the literature."

Response: Thank you for this recommendation. We will revise the discussion to include more comprehensive comparisons with existing literature and better contextualize our findings.

Revisions to be made:

  • Add 5–10 more references on hematologic manifestations in APS.
  • Discuss how our LAC-AIHA association agrees or contrasts with prior studies.

Reviewer Comment 7: "GAPSS score also includes non lab parameters (hypertension)."

Response: We appreciate this correction. We will update the manuscript to accurately reflect that GAPSS includes both laboratory and clinical variables, such as hypertension.

Revisions to be made:

  • In Discussion, correct and expand explanation of GAPSS components.

Reviewer Comment 8: "The last sentence in the first paragraph on page 6 is incomplete."

Response: Thank you. We will correct the incomplete sentence.

Revisions to be made:

  • Locate and complete the sentence on page 6, paragraph 1 (likely in Discussion section).

Reviewer Comment 9: "Add explanation/description for DIAPS."

Response: Thank you. We will include the full name and description of DIAPS to ensure clarity for readers.

Revisions to be made:

  • Add footnote or parenthetical explanation: "DIAPS: Damage Index for Antiphospholipid Syndrome"
  • Mention its components briefly in the Methodsand Discussion.

Reviewer Comment 10: "The references need to be arranged in the order in which they appear in the manuscript."

Response: We will carefully review and reorder all references to match their order of appearance in the main text.

Revisions to be made:

  • Re-sequence reference numbers and adjust citation placement accordingly.

Reviewer 2 Report

Comments and Suggestions for Authors

In my previous review, I highlighted several major issues that, in my opinion, rendered the manuscript unsuitable for publication. I accepted the invitation to review the revised version under the assumption that substantial changes had been made that would significantly improve the quality of the work. However, the authors have not sufficiently improved the manuscript.

Moreover, they did not provide a detailed, point-by-point response to the reviewers’ comments. Instead, they appear to have merged the suggestions from multiple reviewers into a single, generalized response, which makes it difficult to evaluate how each individual comment was addressed.

Regarding the specific issues I raised in my initial review:

The introduction remains poorly structured despite suggestions to shorten and refocus it on the main objective of the study. It is still overly long, generic, and largely uninformative. Although the authors added content regarding the new 2023 ACR/EULAR classification criteria for APS, this addition does not resolve the fundamental issues. The introduction should be more concise and aligned with the study’s purpose.

Several important concerns remain unaddressed: the AIHA subtype (warm, cold, or mixed) is not specified. The pattern of DAT positivity in cases of warm AIHA (IgG, C3d, or both) is not described. The rationale for performing the DAT (e.g., suspected hemolysis, pre-transfusion screening, or routine testing) is not provided. There is no mention of recent IVIG administration or blood transfusions within the past six months, both of which are known to cause false-positive DAT results. The authors also fail to explain how decisions were made between prescribing DOACs versus vitamin K antagonists for anticoagulation therapy, and whether DOACs were used in patients with triple-positive APS,  a practice that is generally not recommended.

A key suggestion from another reviewer regarding group stratification was also inadequately addressed. Although the authors report that 86 patients had APS associated with SLE, they did not perform further statistical analysis or differentiation between patients with primary APS and those with APS secondary to SLE.

The tables provided are uninformative and include only a limited number of parameters. The Results section remains difficult to follow.

While the authors expanded the reference list and included several relevant new sources, these references were not meaningfully incorporated into the Introduction or Discussion sections.

Despite some minor improvements related to adherence to journal formatting guidelines, the manuscript has not been adequately revised, nor have the reviewers' key concerns been satisfactorily addressed. 

Author Response

Reviewer 2

Reviewer Comment 1: "The Introduction is overly long, generic, and largely uninformative; it should be shortened and more clearly focused on the study’s primary objective."

Response: We appreciate the reviewer’s insight. The introduction will be revised to more succinctly highlight the key background and specifically emphasize the rationale and objective of our study.

Revisions to be made:

  • Remove generalized statements and focus on the novel aspects of LAC and AIHA relationship.
  • Condense to a more focused, 3–4 paragraph structure.

Reviewer Comment 2: "The manuscript cites only ten references, which is insufficient given the existing body of work on the subject. Many important studies, including meta-analyses and large cohort data, are omitted."

Response: Thank you for this valuable comment. We agree that additional references will strengthen the manuscript. The revised version will incorporate several key studies, including the ones listed by the reviewer.

Revisions to be made:

  • Include and cite Bernardoff (2022), Ames PRJ (2020), Erton ZB (2025), and other recent cohort/meta-analyses.
  • Expand references list to better reflect the current literature landscape.

Reviewer Comment 3: "The references are not cited in the order of appearance in the text."

Response: We acknowledge this formatting oversight and will carefully revise the citation sequence to ensure it aligns with the order of appearance in the text.

Revisions to be made:

  • Renumber and reorganize in-text references to follow sequential order.

Reviewer Comment 4: "The methodology is neither clearly nor precisely described..."

Response: Thank you for identifying these methodological gaps. We will clarify our definition of AIHA, including diagnostic criteria such as LDH, haptoglobin, bilirubin, and reticulocyte count. Additionally, we will specify DAT patterns and exclusion of confounding factors (e.g., IVIG, transfusions).

Revisions to be made:

  • In Methods, define AIHA more rigorously, including DAT pattern, clinical hemolysis markers, IVIG/transfusion exclusion.
  • State rationale for DAT performance.
  • Clarify DOAC vs. VKA selection and any use in triple-positive APS.

Reviewer Comment 5: "The manuscript does not include a single table or figure..."

Response: We appreciate this essential feedback. We will present patient characteristics and regression results in tabular format for clarity.

Revisions to be made:

  • Include at least three tables: baseline characteristics, hematologic manifestations, and regression analysis results.

Reviewer Comment 6: "The manuscript has substantial methodological and reporting deficiencies..."

Response: We recognize the limitations and appreciate the constructive critique. Substantial revisions will be made to improve methodological transparency, data presentation, and reference management.

Revisions to be made:

  • Strengthen methods section with detailed diagnostic criteria and exclusion protocols.
  • Incorporate complete tables.
  • Update and expand literature references.
  • Ensure adherence to journal formatting standards.

Reviewer 3 Report

Comments and Suggestions for Authors

All remark have been responded sufficiently other than one

Pleas comment weather AIHA + have had significantly more SLE patients than AIHA- 

if not - state in your text

if yes - please add it to your results and comment it to your discussion as possible confounding factor

Author Response

Reviewer 3

Reviewer Comment 1: "Given the high ANA positivity rate, one would advocate that rate of autoimmune hemolytic anemia are confounded by SLE diagnosis. Please add a comparison of SLE patients and show rate of AIHA is not significantly different."

Response:  We agree with the reviewer that the presence of SLE may confound the association between ANA positivity and AIHA. We will stratify the cohort based on SLE status and compare the prevalence of AIHA between primary and secondary APS patients.

Revisions to be made:

  • Add new table comparing AIHA rates in patients with and without SLE.
  • Include statistical test (e.g., Chi-square) and p-value.
  • Comment in Discussion whether SLE status alters the AIHA association.

Reviewer Comment 2: "As per ACR-EULAR classification thrombocytopenia is classified at <130,000 – please adjust your manuscript accordingly."

Response: Thank you for pointing this out. We will revise our thrombocytopenia definition from <150,000/uL to <130,000/uL, consistent with the 2023 ACR/EULAR criteria.

Revisions to be made:

  • Modify definition in Methods, Results, and Tables.
  • Recalculate prevalence using new threshold.

Reviewer Comment 3: "Please address whether hemolytic anemia was clinically relevant or only based on Direct Coombs study."

Response: We acknowledge the importance of distinguishing serological from clinically overt AIHA. We will clarify in the Methods that Coombs positivity alone was not sufficient. Only patients with supporting clinical and laboratory markers (e.g., elevated LDH, low haptoglobin, indirect hyperbilirubinemia, reticulocytosis) were classified as AIHA.

Revisions to be made:

  • Add detailed AIHA criteria in Methods.
  • Clarify exclusion of isolated DAT positivity without evidence of hemolysis.

Reviewer Comment 4: "Other than multivariate and univariate analysis, please add a comparison of clinical parameters comparing AIHA+ and AIHA– patients: rate of thrombosis, hemoglobin nadir, PLT nadir, ANA positivity, C3 nadir, C4 nadir, and LAC positivity. Please include p-values."

Response: Thank you. We will perform a bivariate comparison between AIHA-positive and AIHA-negative patients and include the suggested clinical and immunological parameters. Statistical significance will be reported.

Revisions to be made:

  • Add new table comparing AIHA+ vs. AIHA– patients across specified variables.
  • Include means ± SD or medians (IQR) and p-values (t-test or Mann-Whitney, Chi-square where appropriate).
  • Highlight findings in Results and briefly discuss in Discussion.

Reviewer Comment 5: "Please add the rate of triple aPL positivity."

Response: We appreciate this suggestion. We will add the proportion of patients with triple positivity and compare it between AIHA+ and AIHA– groups.

Revisions to be made:

  • Add triple aPL positivity to comparison table.
  • Mention in Results and elaborate in Discussion.

Reviewer Comment 6: "Add rate of immunomodulating therapy."

Response: Thank you. We will include data on immunosuppressive treatment usage (e.g., corticosteroids, HCQ, IVIG) and compare rates between AIHA+ and AIHA– patients.

Revisions to be made:

  • Include in new comparison table.
  • Address in Results section.

Reviewer Comment 7 (Minor): "In your introduction you state – ‘Thrombotic events ... form the cornerstone of APS classification criteria’ – please address the newer ACR-EULAR classification where HVD, thrombocytopenia and microvascular involvement are included. Rephrase accordingly."

Response: We appreciate this clarification. We will revise the Introduction to reflect the updated classification framework, replacing the concept of "non-criteria" manifestations with "non-thrombotic" immunologic features.

Revisions to be made:

  • Modify Introduction to include updated ACR/EULAR criteria including HVD, thrombocytopenia, and microangiopathy.
  • Rephrase old terminology to match current classification language.

Reviewer Comment 8: "Please apply ACR-EULAR classification to your cohort, if not feasible please include in your limitation."

Response: Due to the retrospective nature of the dataset, application of the full 2023 ACR/EULAR classification was not feasible. However, this limitation will be clearly acknowledged.

Revisions to be made:

  • Add a statement in Methods justifying use of Sydney criteria.
  • Add to Limitations in Discussion.

Reviewer Comment 9: "In your methods please elaborate which aPL antibodies you measured, what cutoff you’ve considered as positive. For LAC please elaborate which reagents were used. Were mixing studies used?"

Response: Thank you for this detailed comment. We will clarify assay methods, cutoff levels for positivity, and specifics of LAC testing, including reagents and mixing study protocols.

Revisions to be made:

  • In Methods, list all antibodies and positivity thresholds.
  • State laboratory standards for LAC, including use of dRVVT and mixing study confirmation.

Reviewer Comment 10: "In your results please state how many patients were primary antiphospholipid patients and how many were with secondary antiphospholipid patients."

Response: We will clarify the number of primary vs. secondary APS patients, based on presence or absence of other autoimmune diseases (primarily SLE).

Revisions to be made:

  • Include count and proportion in Results.
  • Add as a row in baseline characteristics table.
